# 2-Methoxyestradiol Inhibits Radiation-Induced Skin Injuries

**DOI:** 10.3390/ijms23084171

**Published:** 2022-04-10

**Authors:** Ji-Hee Kim, Jae-Kyung Nam, A-Ram Kim, Min-Sik Park, Hae-June Lee, Joonho Park, Joon Kim, Yoon-Jin Lee

**Affiliations:** 1Division of Radiation Biomedical Research, Korea Institute of Radiologic and Medical Sciences, Seoul 01812, Korea; wlgml9054@kirams.re.kr (J.-H.K.); jaek100@kirams.re.kr (J.-K.N.); rhaktyd@naver.com (A.-R.K.); bbrr0044@kirams.re.kr (M.-S.P.); hjlee@kirams.re.kr (H.-J.L.); 2Laboratory of Biochemistry, Division of Life Sciences, Korea University, Seoul 02841, Korea; 3Department of Fine Chemistry, Seoul National University of Science and Technology, Seoul 01811, Korea; jhpark21@seoultech.ac.kr

**Keywords:** 2-Methoxyestradiol, radiation-induced skin injury, HIF 1-α, vascular fibrosis

## Abstract

Radiation-induced skin injury (RISI) is a main side effect of radiotherapy for cancer patients, with vascular damage being a common pathogenesis of acute and chronic RISI. Despite the severity of RISI, there are few treatments for it that are in clinical use. 2-Methoxyestradiol (2-ME) has been reported to regulate the radiation-induced vascular endothelial-to-mesenchymal transition. Thus, we investigated 2-ME as a potent anti-cancer and hypoxia-inducible factor 1 alpha (HIF-1α) inhibitor drug that prevents RISI by targeting HIF-1α. 2-ME treatment prior to and post irradiation inhibited RISI on the skin of C57/BL6 mice. 2-ME also reduced radiation-induced inflammation, skin thickness, and vascular fibrosis. In particular, post-treatment with 2-ME after irradiation repaired the damaged vessels on the irradiated dermal skin, inhibiting endothelial HIF-1α expression. In addition to the increase in vascular density, post-treatment with 2-ME showed fibrotic changes in residual vessels with SMA^+^CD31^+^ on the irradiated skin. Furthermore, 2-ME significantly inhibited fibrotic changes and accumulated DNA damage in irradiated human dermal microvascular endothelial cells. Therefore, we suggest that 2-ME may be a potent therapeutic agent for RISI.

## 1. Introduction

Radiation-induced skin injury (RISI) is a common side effect of radiotherapy for cancer patients. Severe skin injuries may affect the quality of life of cancer patients with radiotherapy [1]. Even though many studies have been conducted on RISIs, treatment methods are limited and currently include steroid and hydrocolloid treatment [2]. Therefore, it is crucial to develop the therapeutic strategy of RISI.

RISIs are classified into acute and chronic injuries. Acute skin injuries are characterized as erythema within hours of irradiation and desquamation within a few weeks, usually within two to six weeks, of radiotherapy, while chronic skin injuries are characterized by fibrosis and necrosis three months to several years after radiotherapy [3,4].

Acute radiation dermatitis within 90 days after ionizing radiation ranges from erythema to desquamation and ulceration. Radiation causes acute damage, with destruction of the basal keratinocytes, and subsequent irradiation induces structural changes in the skin and connective tissues. Radiation fibrosis is a main late effect, characterized by cutaneous induration and retraction, lymphedema, restricted movement, necrosis, and ulceration. The pathogenesis of radiation-induced fibrosis has been thought to be DNA damage, reactive oxygen species, and inflammatory and fibrotic processes [5].

In particular, vascular damage to the dermis and subcutaneous tissue contributes to acute RISI and late fibrosis [6]. The endothelium plays a crucial role in radiation-induced vascular damage [7], and vascular damage can cause radiation-induced late effects. Acute radiation effects include endothelial cell (EC) apoptosis, and late chronic effects include EC senescence and fibrotic changes, which affect vascular homeostasis. Radiation-induced ECs damage can change the balance of pro- and anti-inflammatory, angiogenic, and fibrotic cytokines [8].

The skin has a high concentration of radiosensitive capillaries [6], with a single layer of endothelium. In addition to direct EC death, senescent ECs after radiation mainly contribute to endothelial dysfunction. Radiation-induced fibrosis is known to relate to collagen accumulation and the local expression of transforming growth factor beta (TGF-β) on irradiated regions [9].

Pravastatin, a drug with cholesterol-lowering properties, has been reported to inhibit vascular damage in RISI via regulation of endothelial nitric oxide (eNOS) expression [7]. Quercetin has been reported to reduce dermal fibrotic response to irradiation via collagen accumulation and the expression of TGF-β [10]. Recently, metformin, an anti-diabetic drug, was shown to reduce RISIs by targeting the PI3K-FOXO pathway [11]. In addition, imatinib was reported to inhibit radiation-induced skin fibrosis, reducing transforming growth factor-β (TGF-β) and collagen expression [12]. Despite many studies on the subject, there has been little progress in treatment for RISIs for clinical use.

Endothelial-to-mesenchymal transition (EndMT) predominantly occurs in normal tissue subjected to radiation-induced vascular damage [13,14]. Normal ECs with accumulated DNA damage after irradiation showed phenotypic transition to fibrotic cells [13]. Targeting EndMT has been shown to regulate radiation-induced vascular fibrosis and inhibit radiation-induced pulmonary fibrosis and cardiac damage [13,14]. Vascular EndMT during radiation-induced pulmonary fibrosis (RIPF) development is dependent on the hypoxia-inducible factor 1-alpha (HIF-1α) expression of vascular ECs [14]. 2-Methoxyestradiol (2-ME), which has been approved by the US Food and Drug Administration (FDA) to treat several cancers, has been suggested as an HIF-1α inhibitor [15,16,17]. 2-ME was shown to inhibit radiation-induced EndMT, thereby inhibiting RIPF.

In this study, we investigated the effects of 2-ME on RISIs. 2-ME inhibited HIF-1α-dependent EndMT with reduced accumulated DNA damage after irradiation in human primary dermal ECs. 2-ME efficiently protected against RISIs by reducing skin vascular damage. Furthermore, 7 days after irradiation, 2-ME treatment inhibited vascular injury, which suggests that it could be a potential therapeutic drug for RISI treatment.

## 2. Results

### 2.1. 2-ME Inhibits RISIs

To produce a mouse model of RISIs, 20 Gy was irradiated to the dorsal skin of mice. 2-ME was treated 1 h before irradiation and 24 h after radiation, and the experiment was conducted by dividing the mice into intraperitoneal (I.P.) and oral administration (p.o.) groups. 2-ME was administered once every 2 days a total of six doses. After 21 days of irradiation, the mice were sacrificed, and skin tissues were analyzed (Figure 1a). Hematoxylin and eosin (H&E) and Masson’s Trichrome staining were performed to analyze the degree of skin inflammation and collagen deposition after irradiation. The degree of skin inflammation can be divided into four grades, with grade 1 showing faint or dull erythema, hair removal, dry peeling, and decreased sweating and grade 2 showing tender or bright erythema, mottled wet peeling, and moderate swelling. Grade 3 shows moist peeling and depressed edema, and grade 4 shows ulceration, bleeding, and necrosis [18]. It was confirmed that the level of skin inflammation was significantly decreased in the 2-ME-treated group compared to the untreated group. The thickness of the skin after irradiation was also measured, and it was found that after irradiation, the skin thickness increased in the irradiated group compared to the non-irradiated group. In contrast, it was observed that the change in skin thickness after irradiation was less in the 2-ME treatment group than in the untreated irradiated group (Figure 1a).

It is well known that a single dose of radiation higher than 10 Gy induces severe vascular damages [19]. Immunohistochemistry assay using CD31, an EC marker, showed that the vessel density of irradiated skin significantly decreased compared to that of non-irradiated skin, whereas 1 h before irradiation and 24 h after radiation, 2-ME treatment significantly inhibited the loss of microvessel density and the destruction of microvessels caused by irradiation (Figure 1b). The expression of vascular HIF-1α was increased in the irradiated skin tissues, whereas 2-ME inhibited the expression of vascular HIF-1α (Figure 1b). These results show that 2-ME efficiently inhibits RISIs and regulates radiation-induced vascular damage by reducing vascular HIF-1α expression.

### 2.2. 2-ME Repairs Radiation-Induced Skin Vascular Damage

Next, we investigated whether 2-ME can repair radiation-induced vascular damage. As shown in Appendix A, macroscopic vascular damage occurred 5 days after 17-Gy dorsal skin irradiation. Oral administration of 2-ME was performed once every 2 days starting at 5 and 7 days after irradiation (Figure 2). 2-ME was administered at 10, 30, and 60 mg/kg concentrations. Microvessels destroyed by radiation were prominently restored by post-treatment with 2-ME (Figure 2a, top). H&E and Masson’s Trichrome staining showed that irradiation increased skin inflammation, skin thickness, and collagen deposition, but these were significantly recovered by post-treatment with 2-ME after irradiation (Figure 2a).

As seen in Figure 2a, radiation-decreased microvessel density was significantly restored to the level of that of the non-irradiated group by post-treatment with 2-ME (Figure 2b). Similarly, in the 2-ME post-treated group, the expression level of HIF-1α increased by radiation was decreased (Figure 2b). These results show that 2-ME repaired RISIs by regulating vascular damage. Taken together, we suggest that 2-ME can be a therapeutic agent for RISIs.

### 2.3. 2-ME Regulates Vascular EndMT and Hypoxic Status in RISIs

Previously, we reported that HIF-1α can regulate radiation-induced EndMT [20]. We examined whether residual vascular ECs after irradiation caused HIF-1-dependent EndMT. HIF-1α^+^CD31^+^ vessels were increased in the irradiated tissue, but they were decreased by 2-ME treatment, which indicates that 2-ME regulates the expression of EC-HIF-1α in irradiated skin vessels (Figure 3a). Total HIF-1a + area (%) values in the 30 mg/kg and 60 mg/kg groups were slightly (non-significantly) reduced compared to that in the 10 mk/kg group (Appendix A). Coincidently, radiation-induced EndMT (marked as co-staining with CD31 and αSMA) was decreased by 2-ME treatment both 7 days before irradiation and 24 h after radiation (Figure 3b). Total CD31^+^ area slightly (non-significantly) increased with increasing doses of 2-ME (Appendix A).

As radiation-induced vascular damage enhances the hypoxia of irradiated normal tissue [20], we examined whether 2-ME treatment reduces the hypoxia of irradiated skin tissue (Figure 3c). CA9, a hypoxia marker, was highly positive in ECs and around the cells of irradiated skin, but 2-ME treatment reduced the CA9^+^ area of irradiated skin containing hypoxic ECs (marked as the CA9^+^ podocalyxin^+^ area) (Figure 3c and Appendix A). From these results, we suggest that 2-ME inhibits vascular EndMT and subsequently reduces hypoxia in RISIs by regulating the expression of HIF-1α in ECs.

### 2.4. 2-ME Inhibits Fibrotic Changes in Vascular ECs by Reducing Accumulated DNA Damage in Human Dermal Microvascular ECs (HDMECs)

Through in vivo data, shown in Figure 1b, we found that radiation can cause vascular endothelial death, showing radiation-decreased intensity of CD31^+^ microvessels. We investigated the effects of 2-ME on radiation-induced cell death of HDMECs. Fluorescence activated cell sorting (FACS) analysis showed that 2-ME significantly inhibited radiation-induced cell death in HDMECs (Appendix A).

To determine whether HIF-1α can directly regulate fibrotic changes in ECs, HDMECs were treated with siRNA against HIF-1α. HIF-1α knockdown significantly reduced direct DNA damage (marked as γ-H2AX foci) 1 h after radiation and reduced accumulated DNA damage 72 h after radiation (Figure 4a and Appendix A). In accordance with the regulatory patterns of DNA damage, radiation-induced fibrotic cytoskeletal development was decreased by HIF-1α knockdown using phalloidin staining (Figure 4a). These results show that radiation-induced fibrotic changes in HDECs are dependent on the expression of HIF-1α.

Next, we found that 2-Me treatment 1 h before 20 Gy irradiation efficiently reduced the number of DNA damage foci and subsequent cytoskeleton development in HDMECs (Figure 4b). These results indicate that 2-ME efficiently regulates fibrotic changes in ECs that occur alongside accumulated DNA damage by reducing HIF-1α expression (Appendix A). As shown in Appendix A, 2-ME induced a significant reduction in DNA damage at 30 min, 1 h, and 3 h after irradiation on HUVECs. Radiation increased ATM phosphorylation and DNA-PKcs expression on HDMECs. 2-ME reduction of ATM phosphorylation increased 1 h after radiation, and DNA-PKcs expression was reduced from 10 min after radiation, suggesting that 2-ME can regulate DNA damage through NHEJ repair and HR repair (Appendix A).

## 3. Discussion

RISIs occur in approximately 95% of patients undergoing cancer radiotherapy [21]. In this study, we provide the first evidence that treatment with 2-ME protects against RISIs by reducing vascular damage. Moreover, several days after radiation treatment, 2-ME treatment repaired RISIs, and post-treatment with 2-ME repaired radiation-induced vascular damage, resulting in a density of microvessels and vascular fibrosis similar to those of normal microvessels.

2-ME reduced the foci number of accumulated DNA damage and subsequent fibrotic phenotypic changes in irradiated HDMECs. Interestingly, 2-ME treatment 24 h after radiation reduced accumulated DNA damage. We thus hypothesize that 2-ME treatment would be an efficient therapeutic agent for RISIs.

Radiation-induced microvessel destruction several weeks after radiation, and residual damaged vessels, contribute to tissue fibrosis [22,23]. Here, microvessels were severely damaged 5 days after 20 Gy irradiation of dorsal skin, but they were significantly repaired by 2-ME treatment. Tissue fibrosis is the chronic stage of disordered wound healing and has thus traditionally been regarded as an irreversible process [24]. However, recent studies have shown the reversible nature of organ fibrosis in certain circumstances [25]. Future studies about the repair mechanisms of 2-ME on irradiated fibrotic vessels will be crucial to develop the therapeutic strategy of fibrosis.

Our previous study reported that radiation caused direct DNA damage to ECs, and while it was repaired 24 h after radiation, DNA damage foci remained unrepairable after 48 h in ECs [13]. This persistent DNA damage after irradiation was correlated to fibrotic changes in irradiated ECs. Interestingly, 2-ME can reduce DNA damage in HDMECs in pre- and post-irradiation treatment. We hypothesize that 2-ME can be a radioprotector as well as a mitigator of focal radiation exposure. We are currently studying the application of 2-ME to other radiation-exposed animal models. Concurrently, a recent study of ours has reported that 2-ME inhibits radiation-induced pulmonary fibrosis, which is dependent on HIF-1 expression [14].

2-ME is a metabolite of estradiol with no affinity for estrogen receptors [26]. It is known to inhibit tubulin polymerization via medication such as paclitaxel, colchicine, and paclitaxel [27]. 2-ME, which has been approved by the FDA as an anti-cancer drug, is known to have anti-angiogenic effects and to target HIF-1α [28], and it is used to treat breast, prostate, and ovarian cancer in clinical trials [16,29,30].

Moreover, 2-ME has been reported to enhance the anti-tumor effects of radiation therapy [31,32]. We have also reported that 2-ME enhanced radiation responses of lung adenocarcinoma and inhibited RIPF in preclinical image-guided radiation therapy [30].

The exact mechanism by which 2-ME regulates RISIs remains unclear. Based on our results, we conclude that 2-ME treatment can inhibit vascular destruction and fibrotic development in this RISI model. Furthermore, we suggest the possibility of 2-ME as an mitigator of RISI. We are conducting a further study on the therapeutic possibility and targets of 2-ME in RISIs.

## 4. Materials and Methods

### 4.1. Radiation-Induced Skin Damage Mice Model

All animal experiments were performed with the approval of the Institutional Animal Care and Use Committee of the Korea Institute of Radiological Medicine, in accordance with the Animal Research: Reporting of in vivo Experiments (ARRIVE) guidelines [33].

Irradiation was performed using the X-RAD 320 platform (Precision X-ray, North Branford, CT, USA). C57BL/6 mice skin were irradiated with 17 and 20 Gy using a 4-cm field. 2-ME was injected into the irradiation group. In the pre-treated group, C57BL/6 mice were injected with 2-ME (30 mg/kg, intraperitoneally, oral administration) 1 h before irradiation a total of six times over 3 weeks. In the post-treated group, C57BL/6 mice were treated with 2-ME 1 day (30 mg/kg, intraperitoneally), 5 days (30 mg/kg, oral administration), and 7 days (10, 30, and 60 mg/kg, oral administration) after irradiation. 2-ME (Selleckchem, Houston, TX, USA; #S1233) was dissolved in 2% (*v*/*v*) DMSO (Sigma, Burlington, MA, USA; #D2650) and diluted in 30% (*w*/*v*) PEG 400 (Sigma-Aldrich, St. Louis, MO, USA; #91893) and 1% (*v*/*v*) Tween 80 (Sigma-Aldrich; #P4780).

### 4.2. Histology and Immunohistochemistry Staining

Tissues were fixed in 10% (*v*/*v*) neutral-buffered formalin, embedded in paraffin, and then sectioned. The sections were deparaffinized and stained with Hematoxylin (YD Diagonostics; Yongin-si, Korea; #S251905), Eosin (BIOGNOST, CUB; Zagreb, Croatia; #EOYA-05-OT-1L), and Masson’s Trichrome stain (PolyScience, Warrington, PA, USA; #25088-100). Tissues were stained with H&E for inflammatory analysis and stained with Masson’s Trichrome stain for collagen depositions analysis. For the immunofluorescence and immunohistochemistry staining, the slides were treated with antigen retrieval citrate buffer (Sigma-Aldrich; #C9999) at 95 °C for 30 min and then reacted with 0.3% H_2_O_2_ (Sigma-Aldrich; #216763) for 15 min. For the permeabilization, slides were submerged in phosphate-buffered saline (PBS) containing 0.1% Triton X-100 (PBST) for 15 min and blocked with PBST containing normal horse serum (Vector, Burlingame, CA, USA; #S-2000) for 30 min. The immunofluorescence staining process was the same except for the 0.3% H_2_O_2_ step. We used primary antibodies against CD31(EC marker; 1:200; R&D Systems, Minneapolis, MN, USA; #AF3628), HIF-1α (1:100; Santa Cruz, Dallas, TX, USA #sc-53546), αSMA (fibroblast marker; 1:1000; Sigma-Aldrich; #A5228), γH2AX (DNA damage marker; 1:200; Millipore, Burlington, MA, USA; #05–636), CA9 (hypoxia marker, 1:200, Novus, Littleton, CO, USA; #NB100-417), and phalloidin-Alexa Fluor 488 (1:200; Invitrogen, Waltham, MA, USA; #A12379). An ABC kit (Vector; #PK-6100) and DAB kit (Vector; #PK-4100) were used for immunohistochemistry staining, and the nuclei were stained with hematoxylin. For immunofluorescence, a secondary fluorescent antibody was used, and the nuclei were stained with DAPI (Sigma-Aldrich; #D9542).

Fibrosis grades were determined according to the Ashcroft score, and at least five images per section were acquired to quantify collagen deposition for evaluation using ImageJ software (NIH, Bethesda, MD, USA; http://imagej.net/, accessed on 28 March 2022).

### 4.3. Cell Culture and Treatment

HDMECs (PromoCell, Heidelberg, Germany, DEU; #C-12210) were obtained from PromoCell and cultured in Endothelial Cell Growth Medium MV2 (PromoCell; #C-22022) under a 5% CO_2_ incubator. HDMECs were used within eight passages. Cells were irradiated with BB8000 and then treated with 0.5 μM 2-ME 1 h prior to irradiation.

For HIF-1α silencing experiments, cells were transfected with HIF-1α siRNA (Santa Cruz Biotechnology, TX, USA; #sc-35561) and control siRNA (Santa Cruz; #sc-37007) and used with Lipofectamine 2000 (Invitrogen; #P/N52887). After transfection, the cells were irradiated with 20 Gy and harvested 1, 24, and 72 h after irradiation. The cells were immunofluorescence stained as previous described, with the only difference being that the blocking solution was 1% FBS in PBS.

### 4.4. Flow Cytometer

For the dead cell counting, after PI (Invitrogen; #P3566) staining and FACS analysis, all cells were harvested into FACS tubes and centrifuged at 1300 rpm for 10 min. Next, the supernatant was discarded and cells were washed with PBS. For the flow cytometer settings for the PI staining, the cells were divided into aliquot-stained and non-stained cells. Stained cells used a PI solution (1 μg/mL) and were mixed gently to determine PI fluorescence via FACS analysis.

### 4.5. Immunoblotting

Immunoblotting was performed using antibodies against DNA-PKcs (Santa Cruz; #sc-1552), p-ATM (Millipore, Burlington, MA, USA; 05-740), ATM (Cell Signaling; Danvers, MA, USA; #92356), and β-actin (abcam; Cambridge, UK; #ab8226).

### 4.6. Statistical Analysis

The data were expressed as the mean ± standard deviation (SD) or mean ± standard error of the mean (SEM). One-way analysis of variance (ANOVA) with Tukey’s multiple comparisons test was used for multiple comparisons in GraphPad Prism version 7.0. A *p*-value < 0.05 was considered to indicate statistical significance. The experimenters were blinded to group assignments and outcome assessment. We used ImageJ v1.49 (NIH), R 4.0.4, and Zen 3.2 (Zeiss) for data and image analyses.

## 5. Conclusions

2-ME, an HIF-1α inhibitor that is used as an anti-cancer drug, inhibited RISIs and repaired damaged skin vessels. 2-ME inhibited fibrotic changes in ECs that occurred during radiation-induced vascular fibrosis and reduced both direct DNA damage and unrepairable, persistent DNA damage. Our findings will provide the therapeutic strategy of RISIs, inhibiting both acute cell death and chronic fibrosis of ECs.

## Figures and Tables

**Figure 1 ijms-23-04171-f001:**
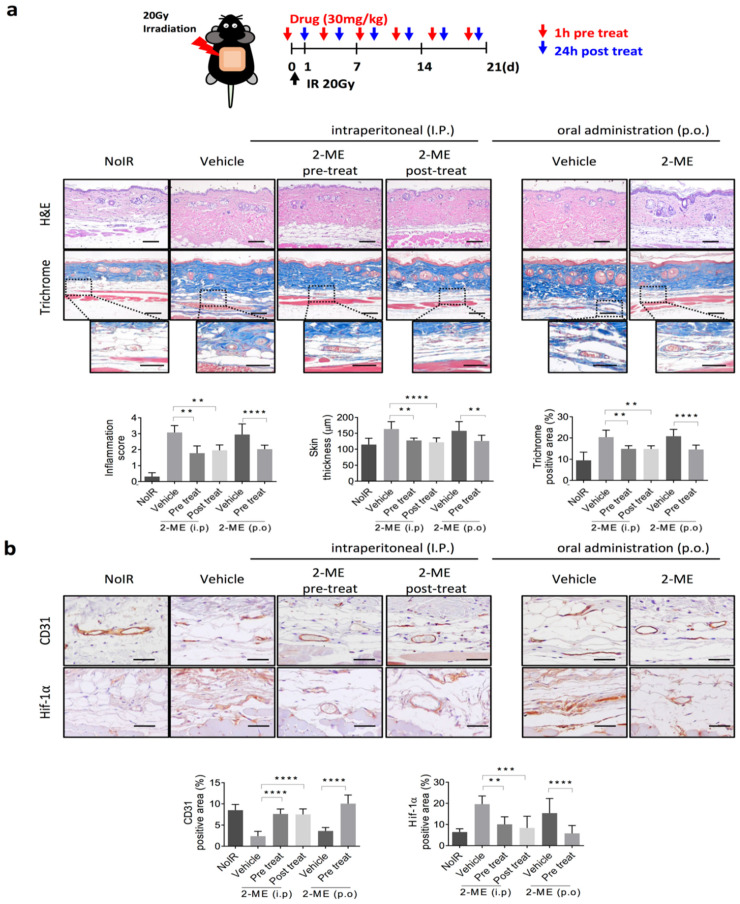
2-ME inhibits radiation-induced skin vascular injuries, perivascular collagen deposition, and upregulated HIF-1α expression. C57BL/6 mice skin were irradiated with 20 Gy using a 4-cm field. Pre-treated mice were injected with 2-ME (30 mg/kg; intraperitoneally or oral administration) 1 h before irradiation a total of 6 times in 3 weeks. Post-treated mice were intraperitoneally injected 24 h after irradiation a total of 6 times in 3 weeks. (**a**) H&E staining and Masson’s Trichrome staining in skin tissues 3 weeks after irradiation with or without 2-ME. Scale bar = 100 μm; scale bar of the cropped images = 50 μm. Quantification of collagen deposition area per field as an average of five fields (magnification, 200×, *n* > 3). Scoring of inflammation, fibrosis grades, and collagen deposition is shown. (**b**) Immunohistochemical detection of CD31 and HIF-1α in skin tissues 3 weeks after irradiation. Scale bar = 50 μm. Quantification of CD31^+^ area and HIF-1α^+^ area per field as an average of five fields (magnification, 200×, *n* > 3). For all graphs, error bars indicate the SD from *n* > 3 biologically independent experiments (one-way ANOVA for multiple comparisons). ** *p* < 0.01, *** *p* < 0.001, and **** *p* < 0.0001.

**Figure 2 ijms-23-04171-f002:**
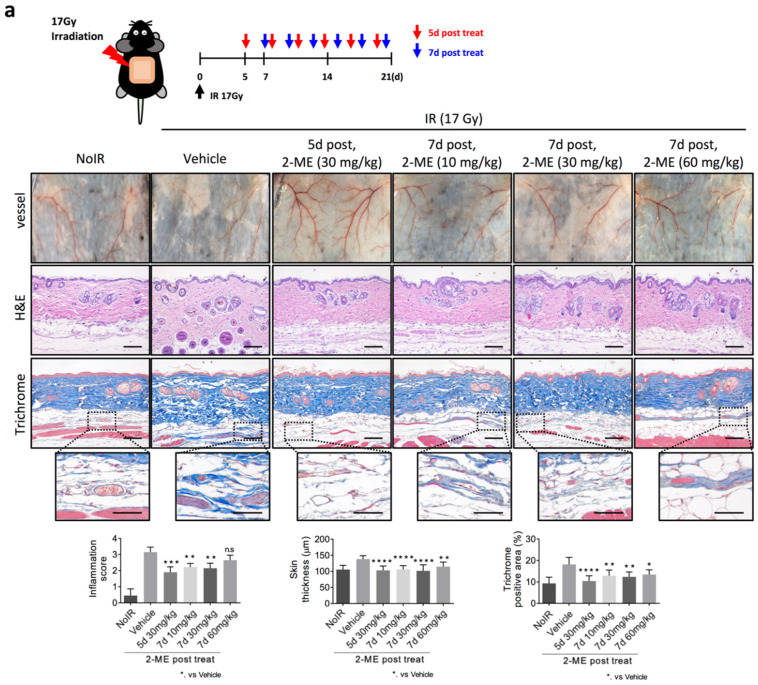
2-ME treatment repairs vascular injuries 5 or 7 days after irradiation, resulting in decreased skin injuries. C57BL/6 mice skin were irradiated with 17 Gy using a 4-cm field. (**a**) Mice were orally injected with 2-ME (30 mg/kg) 5 days and 7 days after irradiation a total of 6 times in 3 weeks. Representative images of skins vessels, H&E staining, and Masson’s Trichrome staining in skin tissues 3 weeks after irradiation with or without 2-ME. Scale bar = 100 μm; scale bar of the cropped images = 50 μm. Quantification of collagen deposition area per field as an average of five fields (magnification, 200×, *n* > 3). Scoring of inflammation, fibrosis grades, and collagen deposition are shown. (**b**) Immunohistochemical detection of CD31 and HIF-1α in skin tissues 3 weeks after irradiation. Scale bar = 50μm. Quantification of CD31^+^ area and HIF-1α^+^ area per field as an average of five fields (magnification, 200×, *n* > 3). For all graphs, error bars indicate the SD from *n* > 3 biologically independent experiments (one-way ANOVA for multiple comparisons). * *p* < 0.05, ** *p* < 0.01, *** *p* < 0.001, and **** *p* < 0.0001.

**Figure 3 ijms-23-04171-f003:**
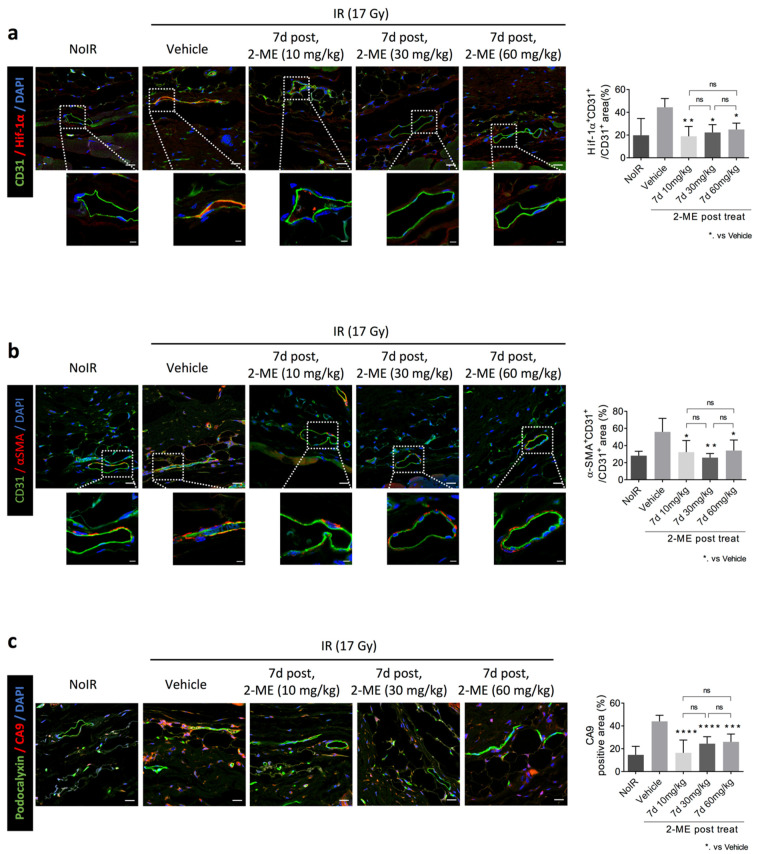
2-ME inhibits irradiation-induced skin vascular EndMT, HIF-1α expression, and CA9 expression. C57BL/6 mice skin was irradiated with 17 Gy using a 4-cm field. Mice were orally injected with 2-ME (30 mg/kg) 5 days and 7 days after irradiation a total of 6 times in 3 weeks. (**a**) Immunofluorescence staining of CD31 (green) and HIF-1α in skin tissues from mice that were non-irradiated and irradiated with or without 2-ME treatment. Scale bar = 20 μm; scale bar of the cropped images = 5 μm. Quantification of the HIF-1α^+^CD31^+^ colocalization area relative to CD31^+^ area as an average of five fields (magnification, 200×, *n* > 5). (**b**) Immunofluorescence staining of CD31 (green) and αSMA in skin tissues from mice that were non-irradiated and irradiated with or without 2-ME treatment. Scale bar = 20 μm; scale bar of the cropped images = 5 μm. Quantification of the HIF-1α^+^CD31^+^ colocalization area relative to CD31^+^ area as an average of five fields (magnification, 200×, *n* > 5). (**c**) Immunofluorescence staining of CA9 and podocalyxin (green) in skin tissues from mice that were non-irradiated and irradiated with or without 2-ME treatment. Scale bar = 20 μm. Quantification of the CA9^+^ area per field as an average of five fields (magnification, 200×, *n* > 5). For all graphs, error bars indicate the SD from *n* > 3 biologically independent experiments (one-way ANOVA for multiple comparisons). * *p* < 0.05, ** *p* < 0.01, *** *p* < 0.001, and **** *p* < 0.0001.

**Figure 4 ijms-23-04171-f004:**
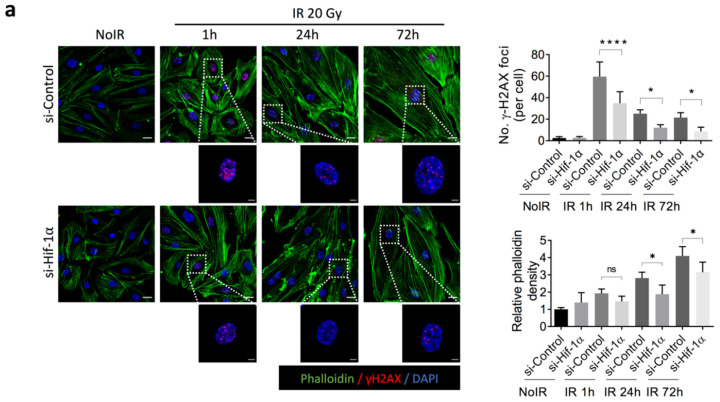
Knockdown of HIF-1α inhibits radiation-induced DNA damage and fibrotic change in HDMECs. (**a**) Immunofluorescence staining of phalloidin and γ-H2AX in HDMECs transfected with HIF-1α siRNA and control siRNA. Transfected HDMECs were irradiated with 20 Gy and harvested at 1 h, 24 h, and 72 h. Scale bar = 20 μm; scale bar of the cropped images = 5 μm. Bar graphs indicate the number of γ-H2AX foci per cell and the relative density of phalloidin (*n* = 4 independent experiments). The average number of foci/cell was determined from five fields (magnification, 200×). For quantification of phalloidin density, error bars represent mean ± SD (one-way ANOVA for multiple comparisons). For quantification of the number of γ-H2AX foci, error bars represent mean ± SD (one-way ANOVA for multiple comparisons). (**b**) Immunofluorescence staining of phalloidin and γ-H2AX in HDMECs treated with 2-ME (0.5 μM) before being irradiated with 20 Gy. (-) represents the treatment of DMSO. The cells were harvested at 1 h, 24 h, and 72 h. Scale bar = 20 μm; scale bar of the cropped images = 5 μm. Bar graphs indicate the number of γ-H2AX foci per cell and the phalloidin density (*n* = 4 independent experiments). The average number of foci/cell was determined from five fields (magnification, 200×). For quantification of phalloidin density, error bars represent mean ± SD (one-way ANOVA for multiple comparisons). For quantification of the number of γ-H2AX foci, error bars represent mean ± SD (one-way ANOVA for multiple comparisons). (**c**) Immunofluorescence staining of phalloidin and γ-H2AX in HDMECs treated with 2-ME (0.5 μM) after 24 h irradiated with 20 Gy. The cells were harvested at 72 h. Scale bar = 20 μm; scale bar of the cropped images = 5 μm. Bar graphs indicate the number of γ-H2AX foci per cell and the relative density of phalloidin (*n* = 4 independent experiments). The average number of foci/cell was determined from five fields (magnification, 200×). For quantification of phalloidin density, error bars represent mean ± SD (one-way ANOVA for multiple comparisons). For quantification of the number of γ-H2AX foci, error bars represent mean ± SD (one-way ANOVA for multiple comparisons). * *p* < 0.05, and **** *p* < 0.0001.

## Data Availability

All other relevant data are available from the corresponding author upon reasonable request.

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
