# Peer review of "2-Methoxyestradiol Inhibits Radiation-Induced Skin Injuries"

_ijms, 2022, doi:10.3390/ijms23084171_

Round 1
Reviewer 1 Report
An interesting original study fcusing on the effects of 2-Methoxyestradiol as a reducing radiation-induced inflammation, skin thickness, and vascular fibrosis drug.
I found the article very interesting, and eligible to be published after minor revisions:
In the introduction, the authors should focus a little more on radiation-induced skin manifestations; here is an article you should read: doi: 10.3390/bioengineering8110153.
In the statistical analysis and throughout the text, please specify the company and location of used products.
Reviewer 2 Report
The study confirmed that, 2-ME, as a potent anti-cancer drug, reduced radiation-induced inflammation, skin thickness, and vascular fibrosis and hence prevented RISI by targeting HIF-1a. In particular, this drug was effective post irradiation treatment and thus has potential clinical application of radiation protection. Although the experimental results are attractive, the molecular mechanism of the function of 2-ME is absent and some of results need to be refined. Therefore, the following key issues should be addressed.
- In the results section, the author tried to demonstrate the effect of 2-ME on radiation-induced skin damage of mice. However, why the irradiation dose in the result 2.1.2 was 20 Gy whereas in the result 2.2.2 was 17 Gy?
- In line 119-122, “H&E and Masson’s Trichrome staining showed that irradiation decreased the level of skin inflammation, skin thickness, and collagen deposition, but these were significantly recovered by post-treatment with 2-ME after irradiation (Figure 2a).” Actually, according to the results shown in Figure 2, it is supposed that irradiation increased the level of skin inflammation, skin thickness and collagen deposition, and that the 2-ME treatment significantly attenuated these responses. The outcomes of Figure 2 and the results (line 119-122) seem to be inconsistent.
- In Figure 3a, since 2-ME acted as an inhibitor of HIF-1α, the expression of HIF-1α should be significantly reduced with increasing doses of 2-ME, that is, the red fluorescence expression should be reduced. However, the red fluorescence in the 30mg/kg and 60mg/kg groups seemed to be stronger than that in the 10mk/kg group in Figure 3a, how to explain this phenomenon?
- In Figure 3b, the expression of CD31 (green fluorescence) should increase significantly with increasing doses of 2-ME, but no gradient increase in green fluorescence was seen in the 10 mg/kg, 30 mg/kg and 60 mg/kg groups. Furthermore, it seemed more convincing to switch a picture of the vehicle group in Figure 3b to make its nuclei number similar to that of the NoIR group.
- In Figure 3 and result 2.3.2, why did CA9 and HIF-1α respond inconsistently to 2-MA? In details, 2-MA acted as an inhibitor of HIF-1α and the increase of 2-MA decreased in HIF-1α expression but increased CA9 expression. What was the reason or mechanism for this?
- This paper suggested that "2-ME promoted RISIs repair by reducing DNA damage through the inhibition of HIF-1a", but the deeply underlying mechanism was not investigated. It has been reported that HIF-1α attenuated DNA damage by regulating Rad52-mediated HR repair (J Nanobiotechnology. 2021 Nov 17;19(1):370), so does 2-ME also act through HR repair in this study? Certainly, the experimental results of this study showed that 2-ME induced a reduction in DNA damage 1 hour after irradiation, and that NHEJ repair mainly works within 1 hour, whereas HR repair generally takes more than 2.5 hour. Therefore, it is possible that 2-ME also works through NHEJ repair. What the effect of 2-ME on the expressions of proteins involved in HR repair, NHEJ repair, and chromatin re-modeling?
Round 2
Reviewer 2 Report
The authors have responsed my questions properly.